# HOXA11-As Promotes Lymph Node Metastasis Through Regulation of IFNL and HMGB Family Genes in Pancreatic Cancer

**DOI:** 10.3390/ijms252312920

**Published:** 2024-11-30

**Authors:** Hayato Nishiyama, Takeshi Niinuma, Hiroshi Kitajima, Kazuya Ishiguro, Eiichiro Yamamoto, Gota Sudo, Hajime Sasaki, Akira Yorozu, Hironori Aoki, Mutsumi Toyota, Masahiro Kai, Hiromu Suzuki

**Affiliations:** 1Department of Molecular Biology, Sapporo Medical University School of Medicine, Sapporo 060-8556, Japan; a19m073@sapmed.ac.jp (H.N.); niinuma@sapmed.ac.jp (T.N.); kitaji@sapmed.ac.jp (H.K.); k05m010@yahoo.co.jp (K.I.); eiichiro@xa3.so-net.ne.jp (E.Y.); yorozuakira0110@gmail.com (A.Y.); hironori_a1123@yahoo.co.jp (H.A.); mutoyota@sapmed.ac.jp (M.T.); kai@sapmed.ac.jp (M.K.); 2Department of Gastroenterology and Hepatology, Sapporo Medical University School of Medicine, Sapporo 060-8543, Japan; gotasudo@gmail.com (G.S.); kiti4989@yahoo.co.jp (H.S.); 3Department of Otolaryngology-Head and Neck Surgery, Sapporo Medical University School of Medicine, Sapporo 060-8543, Japan

**Keywords:** PDAC, lncRNA, lymph node metastasis, IFN-λ, HMGB3

## Abstract

Recent studies have shown that long noncoding RNAs (lncRNAs) play pivotal roles in the development and progression of cancer. In the present study, we aimed to identify lncRNAs associated with lymph node metastasis in pancreatic ductal adenocarcinoma (PDAC). We analyzed data from The Cancer Genome Atlas (TCGA) database to screen for genes overexpressed in primary PDAC tumors with lymph node metastasis. Our screen revealed 740 genes potentially associated with lymph node metastasis, among which were multiple lncRNA genes located in the HOXA locus, including HOXA11-AS. Elevated expression of HOXA11-AS was associated with more advanced tumor stages and shorter overall survival in PDAC patients. HOXA11-AS knockdown suppressed proliferation and migration of PDAC cells. RNA-sequencing analysis revealed that HOXA11-AS knockdown upregulated interferon lambda (IFNL) family genes and downregulated high-mobility group box (HMGB) family genes in PDAC cells. Moreover, HMGB3 knockdown suppressed proliferation and migration by PDAC cells. These results suggest that HOXA11-AS contributes to PDAC progression, at least in part, through regulation of IFNL and HMGB family genes and that HOXA11 AS is a potential therapeutic target in PDAC.

## 1. Introduction

Pancreatic ductal adenocarcinoma (PDAC) is one of the most aggressive and lethal forms of cancer and was ranked as the sixth most common cause of cancer mortality in 2022 [1]. Approximately 80–85% of PDAC patients present with advanced, unresectable disease, which contributes to its lethality [2]. In recent years, improved systemic combination chemotherapy has increasingly extended patient survival, but drug resistance and severe side effects remain significant challenges [3,4]. Lymph node metastasis is strongly associated with the poor prognosis of PDAC patients. Indeed, the 5-year survival rate among patients with and without lymph node metastasis is reportedly 25.0% and 57.1%, respectively, which highlights the clinical significance of lymph node metastasis in PDAC [5].

Most pancreatic intraepithelial neoplasias (PanINs) and 90% of PDACs harbor oncogenic KRAS mutations, underscoring their critical importance in the pathogenesis of PDAC [6,7]. Genetic analyses have shown that loss-of-function alterations in TP53, SMAD4 and CDKN2A following KRAS mutation are essential for PDAC progression [6,7]. Several molecular mechanisms underlying lymph node metastasis have been identified in PDAC. For instance, TIAM1 expression is associated with lymph node metastasis and promotes cell proliferation and invasion by PDAC cells [8]. It has also been reported that high SMAD3 expression is associated with epithelial mesenchymal transition (EMT)-like features and lymph node metastasis [9]. In addition, several microRNAs are reportedly associated with lymph node metastasis in PDAC [10].

Long noncoding RNAs (lncRNAs) are transcripts less than 200 nt in length [11]. Although most lncRNAs are not translated into proteins, they play crucial roles in various biological processes under both normal physiological and cancerous conditions, and a number have been shown to exert oncogenic or tumor-suppressive effects [12]. For example, HOTAIR is upregulated and associated with a poorer prognosis in various malignancies, including breast and gastric cancers and gastrointestinal stromal tumor (GIST) [13,14,15]. On the other hand, MEG3, GAS5 and ANRIL function as tumor suppressors in various types of cancer [16,17,18]. Several lncRNAs, including MALAT1 and TUG1, have been implicated in cancer metastasis [19,20]. MALT1 is upregulated by JMJD2C and activates the β-catenin signaling pathway, which promotes colorectal cancer metastasis [21], while TUG1 is upregulated in osteosarcoma and associates with metastasis by acting as a miR-143-5p sponge [22]. To unravel the molecular mechanisms underlying the progression of PDAC and identify novel therapeutic targets, in the present study, we attempted to identify lncRNAs that promote lymph node metastasis in PDAC.

## 2. Results

### 2.1. Identification of HOXA11-AS Upregulation in PDAC with Lymph Node Metastasis

To identify lncRNAs associated with lymph node metastasis in primary PDAC, we analyzed RNA-sequencing (RNA-seq) data from The Cancer Genome Atlas (TCGA)-PAAD dataset (Figure 1A). By comparing gene expression in primary PDAC with or without lymph node metastasis (metastasis positive, *n* = 49; negative, *n* = 123), we identified 740 genes that were significantly upregulated in tumors with lymph node metastasis (Figure 1A, Appendix A). Gene ontology (GO) analysis revealed that genes associated with digestion, immune response and organelle organization were enriched among the upregulated genes (Figure 1B). Moreover, we noted that nine genes, including seven lncRNAs, were located in the HOXA gene cluster on chromosome 7 (Figure 1C,D). Of the seven lncRNA genes, HOXA11-AS is reportedly upregulated in various types of tumors, though its function is not fully understood [23,24,25]. Data from the CCLE showed abundant expression of HOXA11-AS in multiple PDAC cell lines (Figure 1E). In addition, qRT-PCR analysis confirmed that levels of HOXA11-AS expression were higher in a majority of PDAC cell lines tested than in normal pancreatic tissue (Figure 1F).

### 2.2. Upregulation of HOXA11-AS Is Associated with Malignancy of PDAC

To evaluate the clinical and biological significance of HOXA11-AS in PDAC, we performed further analysis using TCGA-PAAD dataset. Elevated levels of HOXA11-AS expression were associated with lymph node metastasis and advanced T stages in primary PDAC (Figure 2A). By contrast, they were not significantly associated with distant metastasis, likely due to the small sample size (Appendix A). Additionally, higher levels of HOXA11-AS expression were associated with shorter overall survival of PDAC patients (Figure 2B). Upregulation of HOXA11-AS in primary PDAC tumors was then confirmed in an independent cohort (Figure 2C). Gene Set Enrichment Analysis (GSEA) using TCGA dataset revealed that genes associated with the G2/M checkpoint and E2F targets were significantly elevated in primary PDAC tumors with high HOXA11-AS expression (Figure 2D,E). GO analysis using TCGA dataset suggested that expression of genes related to development and morphogenesis were positively associated with HOXA11-AS expression in primary PDAC tumors (Figure 2F).

### 2.3. HOXA11-AS Promotes Proliferation and Migration in PDAC Cells

To confirm the oncogenic function of HOXA11-AS, we next performed a series of knockdown experiments. qRT-PCR analysis showed that levels of HOXA11-AS expression were significantly reduced in PDAC cell lines transfected with siRNAs targeting HOXA11-AS (siHOXA11-AS-1 and siHOXA11-AS-2) (Figure 3A, Appendix A). Cell viability assays revealed that HOXA11 AS knockdown inhibited PDAC cell proliferation (Figure 3B, Appendix A). Transwell assays demonstrated that HOXA11-AS knockdown inhibited PDAC cell migration (Figure 3C). On the other hand, HOXA11-AS knockdown did not significantly reduce the invasiveness of PDAC cells (Figure 3D).

### 2.4. Effects of HOXA11-AS Knockdown on Gene Expression Profiles in PDAC Cells

To further clarify the molecular mechanism by which HOXA11-AS exerts its oncogenic effects, we performed RNA-seq analysis in a PDAC cell line (KP1-NL) transfected with control siRNA or siRNA targeting HOX11-AS. Subsequent RNA-seq analysis showed that 1601 genes were upregulated, while 988 genes were downregulated after HOXA11-AS knockdown in PDAC cells (Figure 4A, Appendix A). GSEA revealed that genes associated with interferon response, TNF-α signaling and JAK-STAT signaling were upregulated by HOXA11-AS knockdown (Figure 4B). GO analysis showed that genes related to immune responses were enriched among the upregulated genes (Figure 4C). Notably, we found that a series of interferon lambda (IFNL) family genes were upregulated, while high-mobility group box (HMGB) family genes were downregulated after HOXA11-AS knockdown (Figure 4A). qRT-PCR analyses confirmed upregulation of IFNL1 and IFNL2 and downregulation of HMGB3 in PDAC cells after HOXA11-AS knockdown (Figure 4D–F).

### 2.5. HMGB3 Promotes Proliferation and Migration in PDAC Cells

Because HMGB3 reportedly exerts oncogenic effects in multiple types of human malignancies, we next analyzed its function in PDAC cells [26,27]. qRT-PCR analysis confirmed successful depletion of HMGB3 expression in PDAC cells transfected with siRNA targeting HMGB3 (Figure 5A). HMGB3 knockdown inhibited proliferation and migration of PDAC cells (Figure 5B,C). GSEA using TCGA dataset showed that expression of genes associated with G2/M checkpoint and E2F targets were elevated in primary PDAC tumors with high HMGB3 expression, which is consistent with the results of GSEA in tumors with high HOXA11-AS expression (Figure 5D,F).

## 3. Discussion

In this study, we demonstrated that HOXA11-AS is upregulated in PDAC with lymph node metastasis and is associated with poor overall survival of PDAC patients. We also found that HOXA11-AS promotes proliferation and migration in PDAC cells. HOXA11-AS is located in the HOXA gene cluster on chromosome 7. Previous studies have demonstrated the clinical utility of HOXA11 immunohistochemistry in diagnosing human malignancies [28]. HOXA11-AS is also reportedly upregulated in various types of human cancer [29]. For instance, increased HOXA11-AS expression is observed in non-small cell lung cancer (NSCLC) and knockdown of HOXA11-AS suppresses proliferation, migration and invasion by NSCLC cells [23]. In primary breast cancer, elevated HOXA11-AS expression is associated with larger tumor size, metastasis and advanced TNM stages [24]. Depletion of HOXA11-AS induces cell cycle arrest and suppresses migration and invasion in breast cancer cells [24]. Upregulation of HOXA11-AS is also associated with poor overall survival in hepatocellular carcinoma (HCC) patients [25]. In HCC cells, HOXA11-AS promotes proliferation by recruiting EZH2 to the promoter region of tumor suppressor genes [25]. Taken together with these observations, our results suggest that upregulation of HOXA11-AS contributes to the progression and malignant phenotype in PDAC.

Our transcriptome analysis revealed that HOXA11-AS suppresses IFNL family genes in PDAC cells. IFN-λs are multifunctional cytokines classified as type III interferons exhibiting high similarity to type I interferons (IFN-α/β) [30]. IFN-λs bind to a heterodimeric receptor complex consisting of IFN-λ receptor 1 (IFNLR1) and interleukin-10 receptor subunit 2 (IL10R2) and activate JAK1 and TYK2, which eventually phosphorylate STAT1 and STAT2 [31]. IFNs, especially IFN-α, have been well studied for their antitumor function and are used to treat various types of cancer, including melanoma, renal cell carcinoma and multiple myeloma [32,33,34]. IFN-λs activate the same pathway as type I IFNs and have been investigated for their anticancer effects. Earlier studies reported that IFN-λs exert tumor-suppressive effects in multiple types of human malignancy, including lung and esophageal cancers and neuroendocrine tumors [35,36,37]. Another study also showed that IFN-λ1 suppresses proliferation and induces apoptosis through upregulation of p21 in PDAC cells [38]. We observed upregulation of multiple IFNL genes in PDAC cells after HOXA11-AS knockdown, suggesting that HOXA11-AS may exert its oncogenic effects through suppression of the IFNL genes.

We also detected downregulation of HMGB family genes in PDAC cells after HOXA11-AS knockdown, which suggests they are downstream targets of HOXA11-AS. Our experimental results and bioinformatical analysis suggest the oncogenic function of HOXA11-AS in PDAC is mediated, at least in part, through upregulation of HMGB3. HMGB1, -2 and -3 share similarities in their amino acid sequences and structures, which consist of two DNA binding domains and a negatively charged C-terminal [39]. HMGB proteins act by modulating chromatin structure and gene transcriptional activity and exert oncogenic effects in various malignancies [40,41,42]. HMGB3 is reportedly upregulated in various human malignancies [43,44,45]. In gastric cancer cells, HMGB3 promotes cell invasion and migration by modulating MMP2 and MMP9 expression [46]. HMGB3 expression also positively correlates with lymph node metastasis in colorectal, urinary bladder and ovarian cancers [26,45,47]. These findings support our hypothesis that HOXA11-AS promotes lymph node metastasis by upregulating HMGB3 in PDAC.

## 4. Materials and Methods

### 4.1. Cell Lines and siRNA Transfection

PDAC cell lines were obtained and cultured in RPMI 1640 medium supplemented with 10% fetal bovine serum (FBS) as described previously. For RNA interference-induced gene knockdown, predesigned Dicer-Substrate Short Interfering RNAs (DsiRNAs) targeting HOXA11-AS and HMGB3 were purchased from Integrated DNA Technologies, Inc. (Coralville, IA, USA). PDAC cells (3 × 10^3^ cells per well in 96-well plates or 1 × 10^5^ cells per well in 12-well plates) were transfected with DsiRNAs (20 nM) or a negative control DsiRNA (20 nM) using Lipofectamine RNAiMAX (Thermo Fisher Scientific, Waltham, MA, USA) according to the manufacturer’s instructions. Sequences of the siRNAs are listed in Appendix A.

### 4.2. Cell Viability Assay

PDAC cells were transfected with DsiRNAs as described above. Cell viability assays were then carried out 24, 48 or 72 h after transfection using a Cell Counting kit-8 (Dojindo, Kumamoto, Japan) according to the manufacturer’s instructions.

### 4.3. Cell Migration and Invasion Assays

Cell migration and invasion assays were performed using transwell chambers as described previously [48]. Briefly, PDAC cells were transfected with DsiRNAs as described above, after which 5 × 10^4^ cells were added to the upper chamber, and RPMI 1640 medium supplemented with 10% FBS was added to the lower chamber. After incubation for 22 h, migrating or invading cells were stained using a Diff quick stain kit (Sysmex, Tokyo, Japan).

### 4.4. RNA Extraction and Quantitative Reverse-Transcription PCR

Total RNA was extracted using a FastGeneTM Basic Kit (Nippon Genetics Co., Ltd., Tokyo, Japan) according to the manufacturer’s instructions. Cells were transfected with DsiRNA as described above, and total RNA was extracted 72 h after transfection. Total RNA from normal pancreatic tissue from a healthy individual was purchased from BioChain (Newark, CA, USA). Single-stranded cDNA was prepared using a PrimeScript RT Reagent Kit with gDNA Eraser Perfect Real Time (TaKaRa Bio, Kusatsu, Japan). qRT-PCR was performed using PowerUp SYBR Green Master Mix (Invitrogen by Thermo Fisher Scientific, Waltham, MA, USA) with a Quant Studio 3 (Applied Biosystems by Thermo Fisher Scientific, Waltham, MA, USA). Relative expression levels of target genes were determined using an endogenous housekeeping gene, beta-2-microglobulin (B2M), as an internal control. Primer sequences are listed in Appendix A.

### 4.5. RNA Sequencing

Total RNA was extracted using a FastGene Basic Kit as described above. Sequencing libraries were prepared using a NEBNext Poly(A) mRNA Magnetic Isolation Module (NEW ENGLAND BioLabs, Ipswich, MA, USA) and a NEBNext Ultra II Directional RNA Library Prep Kit (NEW ENGLAND BioLabs). RNA seq was performed using NovaSeq 6000 (Illumina, San Diego, CA, USA), after which mapping and quantification were performed using the STAR-RSEM pipeline. Data were analyzed using GeneSpring GX version 13 (Agilent Technologies, Santa Clara, CA, USA) and GSEA (Broad Institute, Cambridge, MA, USA). The NCBI SRA accession number for the RNA-seq data is PRJNA1178720.

### 4.6. Data Analysis

RNA-seq data from primary PDAC tumors and normal pancreatic tissues in TCGA dataset and those from PDAC cell lines in the Cancer Cell Line Encyclopedia (CCLE) dataset were obtained from UCSC Xena (http://xena.ucsc.edu/ (accessed on 1 March 2024)) [49]. GSEA and GO analysis using TCGA data were also performed on the UCSC Xena website. Microarray data from primary PDAC tumors and normal pancreatic tissues (GSE55643) were obtained from the Gene Expression Omnibus database (https://www.ncbi.nlm.nih.gov/geo/ (accessed on 1 September 2024)). Data were visualized using the ComplexHeatmap and RCircos package on R 4.3.2.

### 4.7. Statistical Analysis

Quantitative variables were analyzed using Student’s *t*-test or one-way analysis of variance (ANOVA). Survival analysis was performed by constructing Kaplan–Meier curves, which were compared using the log-rank test for two-group comparisons. All data were analyzed using EZR version 1.40 [50].

## 5. Conclusions

In summary, we identified HOXA11-AS as a novel lymph node metastasis-related lncRNA in PDAC. HOXA11-AS may exert its effects by suppressing INFL genes and upregulating HMGB genes in PDAC cells. This suggests HOXA11-AS may be a useful biomarker predictive of lymph node metastasis and a novel therapeutic target in PDAC.

## Figures and Tables

**Figure 1 ijms-25-12920-f001:**
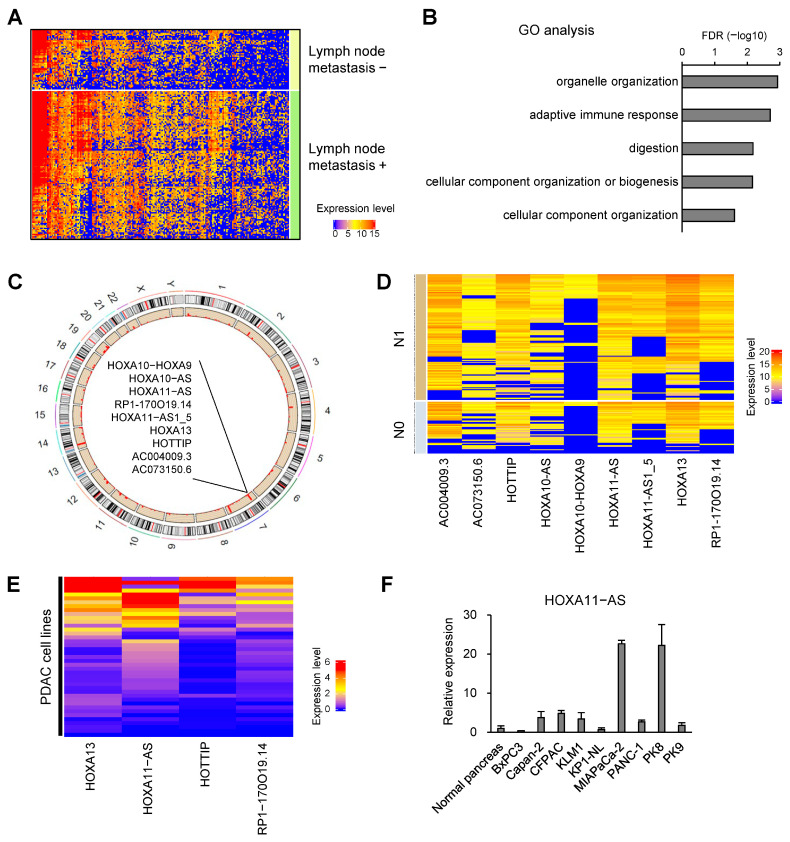
Identification of HOXA11-AS upregulation in PDAC with lymph node metastasis. (**A**) Heatmap showing expression of 740 genes upregulated in PDAC with lymph node metastasis in TCGA dataset. (**B**) GO analysis of genes upregulated in PDAC with lymph node metastasis. (**C**) Circular plot showing the chromosome positions of upregulated genes in PDAC with lymph node metastasis. (**D**) Heatmap showing expression of HOXA cluster genes upregulated in PDAC with lymph node metastasis. (**E**) Heatmap showing expression of HOXA cluster genes in PDAC cell lines in the CCLE dataset. (**F**) qRT-PCR analysis of HOXA11-AS in normal pancreatic tissue and PDAC cell lines (*n* = 3). Error bars represent SDs.

**Figure 2 ijms-25-12920-f002:**
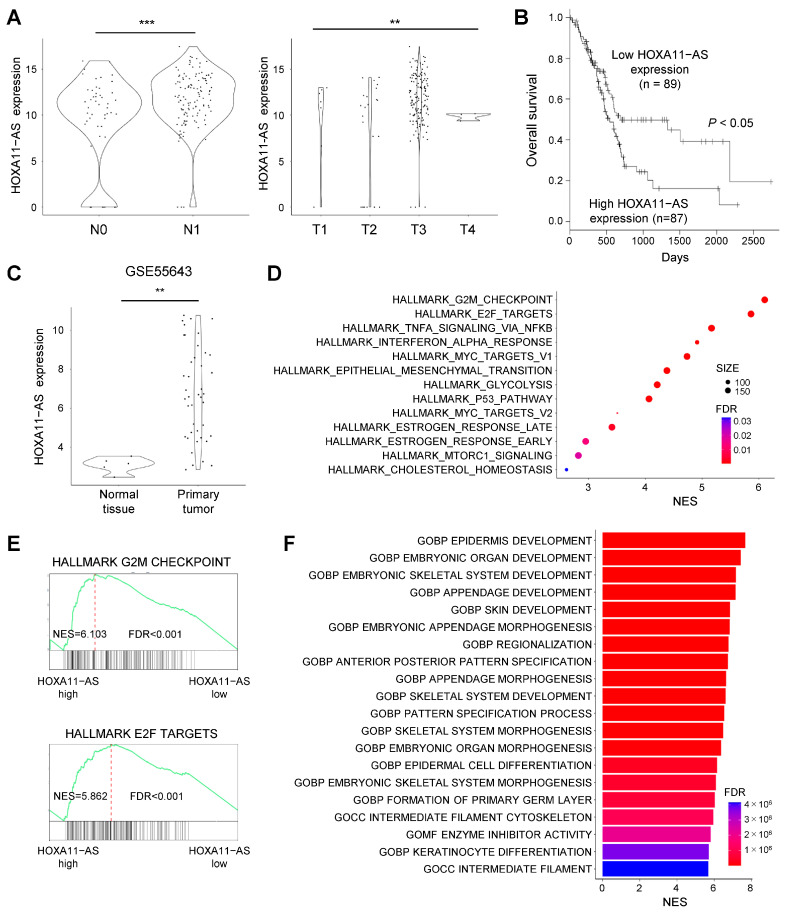
Association between HOXA11-AS expression and clinical and molecular features in primary PDAC. (**A**) Levels of HOXA11-AS expression in primary PDAC tumors with the indicated N factors (left) or T factors (right) in TCGA dataset. (**B**) Kaplan–Meier curves showing the effect of HOXA11-AS expression on survival of PDAC patients (*n* = 176). (**C**) HOXA11-AS expression in normal pancreatic tissue (*n* = 5) and primary PDAC tumors (*n* = 42) in the GSE55643 dataset. (**D**) Summarized results of GSEA of the indicated gene sets using genes upregulated in PDAC with high HOXA11-AS expression in TCGA dataset. NES, normalized enrichment score; FDR, false discovery rate. (**E**) Results of GSEA of the indicated gene sets in PDAC with high HOXA11-AS expression. (**F**) Summarized results of GO analysis using genes upregulated in PDAC with high HOXA11-AS expression in TCGA dataset. ** *p* < 0.01, *** *p* < 0.001.

**Figure 3 ijms-25-12920-f003:**
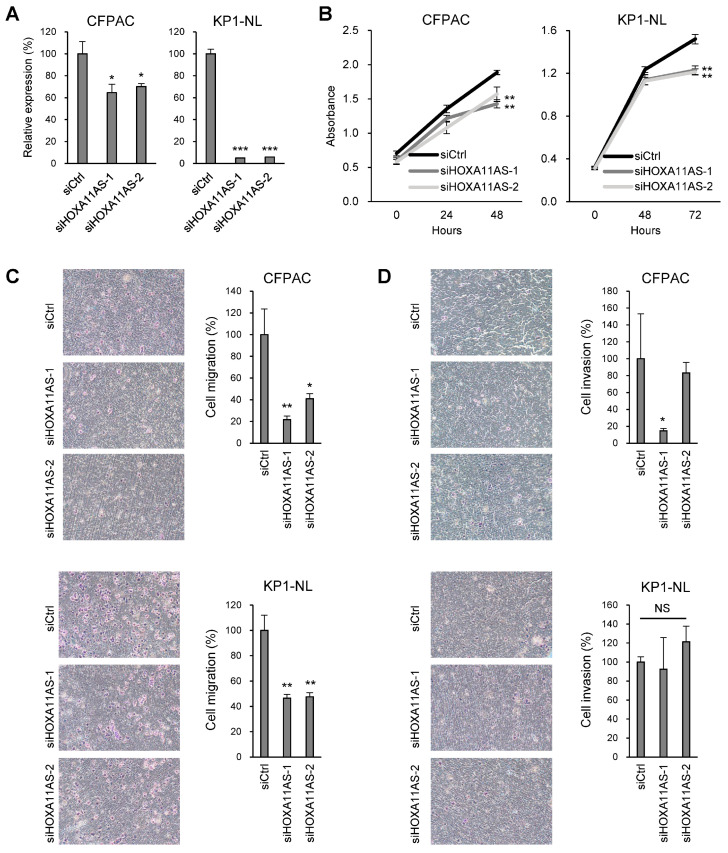
Functional analysis of HOXA11-AS in PDAC cells. (**A**) qRT-PCR analysis of HOXA11-AS in PDAC cells transfected with a control siRNA or siRNAs targeting HOXA11-AS (*n* = 3). (**B**) Cell viability assays with PDAC cells transfected with the indicated siRNAs (*n* = 6). (**C**,**D**) Cell migration (**C**) and invasion (**D**) assays with PDAC cells transfected with the indicated siRNAs. Representative results are shown on the left, summarized results on the right (*n* = 3). Error bars represent SDs. * *p* < 0.05, ** *p* < 0.01, *** *p* < 0.001, NS, not significant.

**Figure 4 ijms-25-12920-f004:**
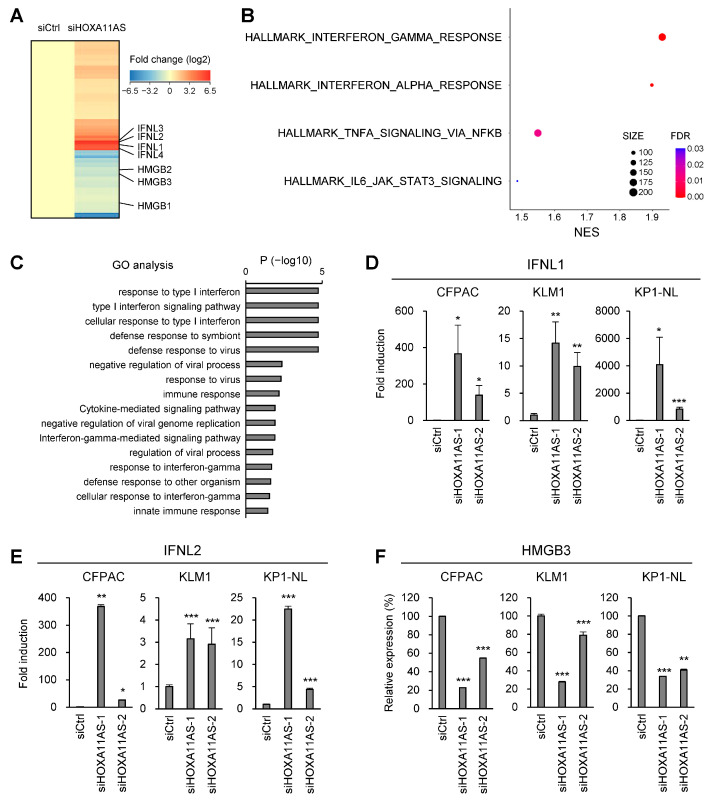
HOXA11-AS regulates IFNL and HMGB family genes in PDAC cells. (**A**) Results of RNA-seq in KP1-NL cells transfected with the indicated siRNAs. Shown is a heatmap of genes whose expression was altered (>1.5-fold) by HOXA11-AS knockdown. Representative genes are indicated on the right. (**B**) Summarized results of GSEA of the indicated gene sets using genes upregulated by HOXA11 AS knockdown. (**C**) GO analysis of genes upregulated by HOXA11-AS knockdown. (**D**–**F**) qRT-PCR analysis of IFNL1 (**D**), IFNL2 (**E**) and HMGB3 (**F**) in the indicated PDAC cell lines transfected with the indicated siRNAs (*n* = 3). Error bars represent SDs. * *p* < 0.05, ** *p* < 0.01, *** *p* < 0.001.

**Figure 5 ijms-25-12920-f005:**
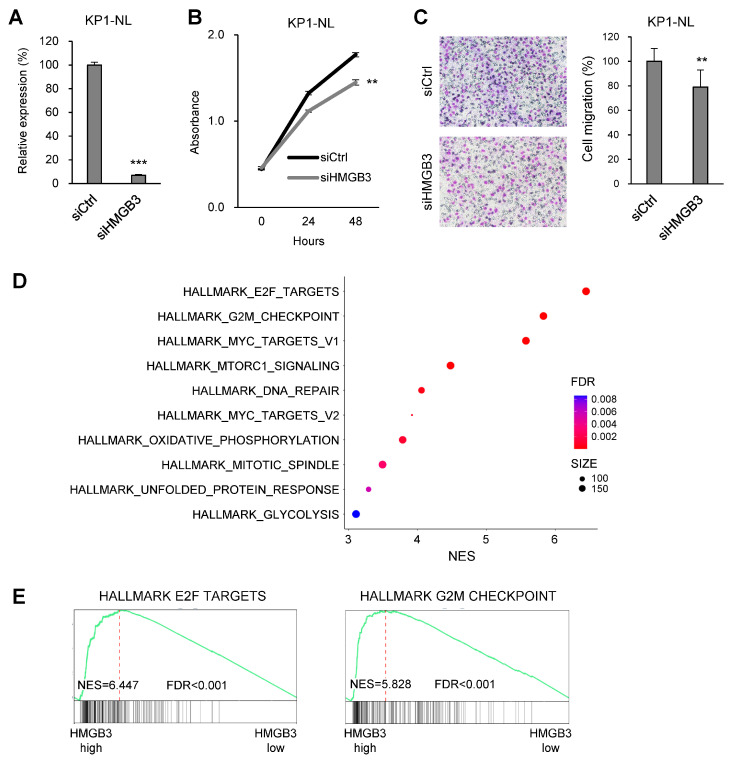
Functional analysis of HMGB3 in PDAC cells. (**A**) qRT-PCR analysis of HMGB3 in KP1-NL cells transfected with a control siRNA or siRNA targeting HMGB3 (*n* = 3). (**B**) Cell viability assays with KP1-NL cells transfected with the indicated siRNAs (*n* = 6). (**C**) Cell migration assays with KP1-NL cells transfected with the indicated siRNAs. Representative results are shown on the left, summarized results on the right (*n* = 3). (**D**) Summarized results of GSEA of indicated gene sets using genes upregulated in PDAC with high HMGB3 expression in TCGA dataset. (**E**) Results of GSEA of the indicated gene sets in PDAC with high HMGB3 expression. Error bars represent SDs. ** *p* < 0.01, *** *p* < 0.001.

## Data Availability

The NCBI SRA accession number for the RNA-seq data is PRJNA1178720.

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
