# Peer review of "HOXA11-As Promotes Lymph Node Metastasis Through Regulation of IFNL and HMGB Family Genes in Pancreatic Cancer"

_ijms, 2024, doi:10.3390/ijms252312920_

Round 1
Reviewer 1 Report
Comments and Suggestions for Authors
The authors present a very nice paper on PDAC. PDAC is highly letal, especially in cases with lymph node metastases
The study is solid and rational, with good results and discussion
Figures are of good quality and references are update.
As improvement, I sugest expandig the discussion with the possibility of immunohistochemistry for HOXA, especially in biopsies, allowing to predict patients with low stage tumor for prompt surgery. This IHC is already availabe and used in other circunstances (https://pubmed.ncbi.nlm.nih.gov/34844098/)
Author Response
Comment 1: The authors present a very nice paper on PDAC. PDAC is highly letal, especially in cases with lymph node metastases. The study is solid and rational, with good results and discussion. Figures are of good quality and references are update.
Response 1: We thank the reviewer for the very positive comments on our manuscript.
Comment 2: As improvement, I sugest expandig the discussion with the possibility of immunohistochemistry for HOXA, especially in biopsies, allowing to predict patients with low stage tumor for prompt surgery. This IHC is already availabe and used in other circunstances (https://pubmed.ncbi.nlm.nih.gov/34844098/)
Response 2: As suggested the reviewer, we now cite the manuscript (PMID 34844098) and expanded our discussion.
Reviewer 2 Report
Comments and Suggestions for Authors
Well-designed study to highlight the role of HOXA11-AS in promoting lymph node metastasis through regulation of IFNL and HMGB family genes in pancreatic cancer. The data included in the manuscript support the conclusion. However, including the data using a plasmid to increase the expression of HOXA11-AS and then including the results of migration, proliferation, colony formation assay, and wound healing assay will be great. Also, colony formation assay results should be included using siRNA. Also, please include the IHC images of lymph nodes with a positive staining for HOXA11-AS, HMGB3, and IFNL from protein atlas if possible.
Author Response
Comment 1: Well-designed study to highlight the role of HOXA11-AS in promoting lymph node metastasis through regulation of IFNL and HMGB family genes in pancreatic cancer. The data included in the manuscript support the conclusion.
Response 1: We thank the reviewer for the very positive comments on our manuscript.
Comment 2: However, including the data using a plasmid to increase the expression of HOXA11-AS and then including the results of migration, proliferation, colony formation assay, and wound healing assay will be great. Also, colony formation assay results should be included using siRNA.
Response 2: We agree that additional experiments using a plasmid vector and additional colony formation assays would further strengthen our study. Because the time until deadline of re-submission is limited, we would like to address these issues in our future study.
Comment 3: Also, please include the IHC images of lymph nodes with a positive staining for HOXA11-AS, HMGB3, and IFNL from protein atlas if possible.
Response 3: We thank the reviewer for the very helpful comment. We checked the protein atlas database, but lymph node samples of pancreatic cancer patients are not available. More importantly, because HOXA11-AS is a non-coding RNA, it is impossible to perform immunohistochemistry of HOXA11-AS.